# Incorporation of Tumor-Free Distance and Other Alternative Ultrasound Biomarkers into a Myometrial Invasion-Based Model Better Predicts Lymph Node Metastasis in Endometrial Cancer: Evidence and Future Prospects

**DOI:** 10.3390/diagnostics12112604

**Published:** 2022-10-27

**Authors:** Marcin Liro, Marcin Śniadecki, Ewa Wycinka, Szymon Wojtylak, Michał Brzeziński, Joanna Jastrzębska, Dariusz Wydra

**Affiliations:** 1Department of Gynecology and Obstetrics, Medical University of Gdańsk, 80-210 Gdańsk, Poland; 2Department of Statistics, Faculty of Management, Gdańsk University, 80-309 Gdańsk, Poland; 3Department of Pathology, Medical University of Gdańsk, 80-210 Gdańsk, Poland; 4Department of Gynecologic Oncology, PCK Marine Hospital in Gdynia, 81-519 Gdynia, Poland

**Keywords:** endometrial cancer, ultrasound, lymph nodes, staging, metastases, biomarkers, model

## Abstract

Myometrial invasion (MI) is a parameter currently used in transvaginal ultrasound (TVS) in endometrial cancer (EC) to determine local staging; however, without molecular diagnostics, it is insufficient for the selection of high-risk cases, i.e., those with a high risk of lymph node metastases (LNM). The study’s objective was to answer the question of which TVS markers, or their combination, reflecting the molecular changes in EC, can improve the prediction of LNM. Methods: The TVS examination was performed on 116 consecutive EC patients included in this prospective study. The results from the final histopathology were a reference standard. Univariate and multivariate logistic models of analyzed TVS biomarkers (tumor [T] size, T area [AREA], T volume [SPE-VOL], MI, T-free distance to serosa [TFD], endo-myometrial irregularity, [EMIR], cervical stromal involvement, CSI) were evaluated to assess the relative accuracy of the possible LNM predictors., Spline functions were applied to avoid a potential bias in assuming linear relations between LNM and continuous predictors. Calculations were made in R using libraries *splines*, *glmulti*, and *pROC*. Results: LNM was found in 20 out of the 116 (17%) patients. In univariate analysis, only uMI, EMIR, uCSI and uTFD were significant predictors of LNM. The accuracy was 0.707 (AUC 0.684, 95% CI 0.568–0.801) for uMI (*p* < 0.01), 0.672 (AUC 0.664, 95% CI 0.547–0.781) for EMIR (*p* < 0.01), 0.776 (AUC 0.647, 95% CI 0.529–0.765) for uCSI (*p* < 0.01), and 0.638 (AUC 0.683, 95% CI 0.563–0.803) for uTFD (*p* < 0.05). The cut-off value for uTFD was 5.2 mm. However, AREA and VOL revealed a significant relationship by nonlinear analysis as well. Among all possible multivariate models, the one comprising interactions of splines of uTFD with uMI and splines of SPE-VOL with uCSI showed the most usefulness. Accuracy was 0.802 (AUC 0.791, 95% CI 0.673–0.91) Conclusions: A combination of uTFD for patients with uMI > 50%, and SPE-VOL for patients with uCSI, allows for the most accurate prediction of LNM in EC, rather than uMI alone.

## 1. Introduction

Although the histological and molecular features of a tumor facilitate a differential prognosis of endometrial cancer (EC), ultrasonographic (US) examination is a fundamental part of an EC patient’s work-up [1,2]. Ultrasound in EC has value for several reasons; it is useful to determine the pre-operative tumor extension, it is relatively easy to use and widely available in many centers, and because gynecological ultrasound training begins with the assessment of the uterus and endometrium. So far, most studies have been devoted to analyzing the prognostic and predictive utility of US for uterine infiltration, namely, invasion depth in relation to full myometrial thickness, expressed as either greater than, equal to, or less than 50% [3,4]. However, myometrial invasion measured by ultrasonography (uMI) has its drawbacks, such as the irregularity of the endomyometrial junction, difficulty in assessing cancer invasion by adenomyosis, or a variety of unusual invasion patterns of EC [5,6,7]. Therefore, new biomarkers are sought that can either serve as an alternative or a complement to uMI. The common denominator and reference point for assessing the usefulness of ultrasound biomarkers should be whether they can predict lymph node metastases (LNM), which are a critical (but not the only) characteristic in the high-risk EC group [8,9]. So far, several biomarkers have been mentioned in the literature: tumor size [10], tumor surface area [11], tumor volume [12], tumor-free distance (uTFD) [13], myometrial invasion (uMI ≥ 50%) [14], endomyometrial irregularity (EMIR) [15], and cervical invasion (uCSI) [16]. However, simultaneous analysis of all these biomarkers has not yet been undertaken with the power to create a model for predicting EC LNM. Our study aims to develop such a model that is as simple to use as uMI, but more effective at predicting the risk of nodal metastases following the diagnosis of EC. The social context of US is also important as it is a common imaging modality in gynecology. This mainly applies to EC, which is one of the most common cancers in women worldwide [17].

## 2. Materials and Methods

The authors acknowledge that the portion of the results section concerning the uTFD parameter and the measurement method discussed as a replacement for the standard uMI parameter have already been published in “Diagnostics” [18]. The present article, which is a continuation of the previous work, deals with all possible biomarkers more broadly as models, as discussed below, to find which is best for predicting lymph node status. The study was approved by The Research Ethics Committee of the Medical University of Gdansk, and each patient voluntarily gave their written informed consent to participate in the study, on the understanding that therapeutic decisions, except for uMI, were not dependent on the results of these measurements. The study was conducted in accordance with the ethical principles for medical research of the Declaration of Helsinki [19]. The study uses the Standards for Reporting Diagnostic accuracy studies (STARD) 2015 guidelines for reporting results [20].

### 2.1. Study Design and Participants

In addition to the data analyses of our previously published work [18] (Table 1), and with the same 116 patients recruited between January 2011 and November 2012 for ultrasound analysis of their endometrial cancer, in the present study, we have considered all the parameters that we were collecting at that time, to provide more complete data. In the present study, the idea was to use the prospective data on ultrasound biomarkers we collected to determine which model, consisting of no more than two factors, would be better than using uMI alone. In Figure 1, we have presented the study’s inclusion and exclusion criteria for patients, which are the same as those used in the previously published study [18]. Briefly, the inclusion criteria were as follows: adult patients with EC confirmed by dilation and curettage (D&C) or hysteroscopy prior to surgery, referred from either our outpatient or external gynecological care units or other hospitals. Patients with myoma or/and adenomyosis and FIGO stage IV cancers were excluded from the study.

### 2.2. Ultrasound Examination

2D transvaginal ultrasound (TVS) was performed using the Philips HD7 (Koninklijke Philips N.V.) with a transvaginal transducer (frequency 6–12 MHz, depth of imaging ranging from 10 to 12 cm, gray map 256 (8 bits). The following ultrasound (u) markers were analyzed: tumor size (T), tumor area (AREA), volume (VOL), myometrial invasion (MI), tumor-free distance (TFD), endo-myometrial irregularity (EMIR), and cervical stromal invasion (CSI). T feature was measured as the largest dimension of the tumor at either the frontal, sagittal, or transverse planes; AREA was assessed at the largest dimension of the tumor (most often in or near the sagittal plane). In turn, VOL was measured planimetrically, by calculating the three dimensions of the suspected echogenic structure, according to the formula: π/6 × d1 × d2 × d3 (where “d” is the dimension). We made three measurements for both uAREA and uSPE-VOL and used the highest value in each parameter to minimize the calculation error. In the same three planes, uTFD measurements were made subjectively in the most locally advanced part of the tumor; and the shortest distance from the forehead of the infiltration to the serosa surface was taken into consideration. The ultrasound MI was measured by subtracting the tumor thickness (perpendicular to the long axis) from the distance between the endometrium–myometrium interface to the serosa. Ultrasonographically-measured EMIR was examined by accentuating the endometrial junction— if this border had been breached (it could not be traced), the trait was deemed positive. The uCSI measured by ultrasound was defined as the absence of the outline of the tumor, at least in the inner orifice of the cervix. Figure 2 (panel) shows an example of measurement-taking. All measurements were made using a tension-free technique, so as not to compress the tissues and thus to avoid distortion of the results. The ultrasonographer was aware of the primary pathological result. The staging was determined preoperatively based on TVUS according to the 2009 International Federation of Gynecology and Obstetrics (FIGO) classification system [21].

### 2.3. Surgery, Including Lymph Nodes Procedure

We described the surgical procedures in our previous publication [18]. The types of surgery included were simple hysterectomy and bilateral salpingo-oophorectomy with sentinel lymph node biopsy (SLNB) or pelvic lymphadenectomy in endometrioid carcinomas, and total hysterectomy with salpingo-oophorectomy with pelvic and para-aortic lymphadenectomy (in those patients with risk factors for lymph node recurrence and metastasis: serous EC, grade 3 endometrioid subtype, uMI equal to or higher than 50%, cervical involvement). Sentinel lymph node biopsy was performed in patients with contraindications for extensive lymph node surgery (e.g., poor general condition or comorbidities).

The SLNB concept that we used was based on combining the Tc99m-nanocolloid applied to the ectocervix mucosa before the skin incision and the intraoperative injection of blue dye to the subserosa of the uterine fundus. During the procedure, we assessed node colour and radiotracer uptake; when blue staining occurred and/or uptake ten times the background level, they were determined to be SLN-positive.

### 2.4. Histopathology

The pathologist received the ultrasound results blind. All the results received from external sites were subject to verification internally. That is, for each sample received, each external institution’s blocks (for instance, histological slides) underwent pathologist processing and verification by a specialist in our facility. The excised lymph nodes were subject to routine histopathological treatment (reference standard) [18].

### 2.5. Statistical Analysis

Univariate logit models for continuous predictors were evaluated twice (raw predictors and cubic splines of predictors). Univariate logit models were also evaluated for qualitative predictors (uT, uMI, EMIR, uCSI,). To build the multivariate models, all the possible combinations of covariates, as well as the interactions between them, were considered. Using the *glmulti* package in R, more than 450 models were estimated. The Akaike Information Criterion (AIC) was used to select the best multivariate model and to omit overestimation [24]. The discrimination ability of models was assessed with the use of a receiver operating characteristic curve (ROC) and the area under it (AUC). Accuracies were calculated for points of predictors that maximize Youden’s index. The likelihood ratio test (LRT) was used as a global test for models. In logit models, to avoid a risk of overfitting, a minimum of 10 outcome events per predictor variable should be used. This rule was established in simulation studies [25]. Thus, having 20 outcome events, the model with up to two predictors can be specified.

## 3. Results

In Table 1, we presented the basic characteristics of the patients in our study. We recalled the results of a previous study that included the same group of patients [18].

Table 2 summarises the values of seven ultrasound variables in the study group.

Univariate models showed the influence of each of the ultrasound predictors on the risk of lymph node metastases, namely, the influence of uTFD (C model), uSPE-VOL and AREA (nonlinearly, D and E models), uMI (G model), EMIR (H model), uCSI (at the limit of statistical significance), and size (model J). Models A and B were irrelevant, i.e., there is no linear influence of uSPE-VOL and uAREA on LNM. Among univariate models, the highest accuracy was achieved by the uCSI model, with a result of 77.6% (Table 3). The multivariate model (K) includes two interactions; the first uMI with bs(uTFD) and the second uCSI with bs(SPE-VOL). The effect of uTFD and uSPE-VOL on LNM does not appear to be linear. This nonlinearity can be modeled using splines. They equalize the values of the variable in successive intervals of a fixed length with the use of polynomials. The parameters of the polynomials are estimated to obtain the smoothness of the polynomial connections. The curve that is glued together is called a spline. The B-spline (bs) technique is most often used. According to Akaike information criterion, the K model is the best model (AIC = 94.81). For comparison, the best one-way model was uMI (AIC = 101.17). The multivariate (K) model also achieved the highest accuracy (80%) in predicting metastasis. The comparison of the ROC curves of the univariate and multivariate models, proving the latter’s superiority, is presented in the graph in Figure 3.

As SPE-VOL increases to 40 cm^−3^, the probability of LNM increases and then decreases (Appendix A); in the case of AREA, it is an increase of up to 18 cm^−2^. As the tumor surface continues to enlarge on ultrasound, the risk of LNM decreases.

For the high-risk group, no single-factor model was significant in LNM prediction. Therefore, we tried to find a model where one predictor would show the interaction of the variables. The uCSI:uMI model turned out to be the best (AIC 45.19, ACC 71%) (Table 4).

For the low-risk group, the uMI:EMIR interaction is important, but too few metastases were recorded in this large group of patients to build a reliable model (Appendix A).

## 4. Discussion

Until now, uMI has been thought to be the only reliable marker in the prediction of lymph node metastases in EC. Our study has shown that there are other ultrasound markers, such as uTFD, uCSI or uSPE-VOL, that add predictive value to considering uMI alone. The dependencies, however, are not linear and require the use of nonlinear mathematical models to demonstrate these relationships.

Two ultrasound parameters, uMI and uCSI, were demonstrated to be the best single predictors of LNM in EC. However, the multivariate model, consisting of uMI:bs(uTFD) and uCSI:bs(uSPE-VOL) pairs, showed higher accuracy than univariate models. The parameters uMI and uCSI are well-known predictors of LNM, while uTFD and uSPE-VOL are new predictors. The latter two seem to increase the predictive power of “classical” parameters. In the clinical anatomical sense, the first, uTFD, shows cancer access to blood vessels with a cut-off value of 5 mm from the serosa, and perhaps this is already evidence of the infiltration of the tumor to the lymph vessels. The second ultrasound biomarker, uSPE-VOL, showed the tumor–uterus relationship, whereas large tumors, i.e., over 40 cm^3^, showed more mildly invasive features. This latter can be explained by the length of their growth until diagnosis, especially since we do not often observe LNM in them. The opposite may be true for rapidly growing, aggressive (possibly metastatic) tumors, which are found to be smaller at diagnosis. The uMI:uCSI model best defined the biological aggressiveness of cancer (high-risk tumors), which can be explained by the fact that both markers in this model are predictors of invasion.

Our study has several limitations. The first limitation is the low number of cases with positive lymph nodes in our sample (20/116). Therefore, it was impossible to divide the risk groups into more than two categories (high risk/low risk), with the caution that histological and ultrasound features, but not molecular features, were considered. In addition, most of the low-risk group did not undergo a full lymphadenectomy, which (although consistent with the treatment guidelines) did not allow for a comprehensive comparison with the high-risk group in which full lymphadenectomy had been performed [1,26]. The third limitation was the influence of the diagnostic procedures used prior to the diagnosis of EC (D&C, hysteroscopy), affecting the accuracy of imaging prior to surgery. For example, the results of volumetric-based biomarkers (such as uSPE-VOL) must be taken with caution, because during invasive procedures such as D&C and endoscopic techniques, some tissues may be lost before the preoperative ultrasound assessment is undertaken. The fourth limitation is that biomarkers indicating continuity disturbance of the boundary between the tumor and healthy tissue (such as uMI and EMIR) can also indicate the same critical phenomenon, such as the invasion process. Thus, as one marker overlaps the other, the clarity of the modelling is reduced. On the other hand, the difference between uMI and EMIR is that uEMIR is a zero-one feature, and uMI is a semi-quantitative biomarker. The next limitation is that we did not include adenomyosis and myoma patients. This is because uEMIR cannot be properly assessed in such cases, due to the fact that the intraepithelial zone may be poorly reflected in ultrasound if some disease of the uterine muscle is present. Lastly, we did not incorporate cancer grading in our ultrasound models to ensure the correct methodology for the ultrasound trial.

It is assumed that ultrasound with a vaginal transducer is a diagnostic tool that does not permit direct visualization of lymph node metastases [27]. More advanced imaging (i.e., computed tomography, magnetic resonance imaging) can detect LNM directly, but it cannot detect micrometastases. This means that at the point of EC diagnosis those tests do not have a significant advantage over TVUS examination [28,29].

Ultrasound-measured MI is a guide for deciding whether to perform lymphadenectomy prior to surgical intervention. This biomarker has proved to be a decisive factor when determining the scope of operation in cases of potentially high-risk tumors. Being imprecise, uMI seems to be a key biomarker for predicting EC cases as high risk but is not a sufficient parameter for determining LNM risk [30,31,32]. In general, uMI is associated with other parameters besides uCSI, such as tumor histology. There is little data on ultrasound-only parameter models’ usefulness in predicting metastases [33,34,35]. Our study is the only one that covers ultrasound-only models, irrespective of grade or tumor type.

It is known, however, that histopathological, and, nowadays, molecular examination determines the further treatment of patients with EC. The question is how widespread these approaches are and what characterizes the “false low-risk” group within the “low-risk” group. In our study, most cases were “low risk”, and we performed a separate analysis for “low risk” ECs. In this group of 6 “false low risk” cases, the risk was higher than it would appear from their classification. There was no stromal infiltration in this subgroup at all; 5/6 had uTFD up to 5.2 mm, 4/6 had EMIR, and 3/6 had uMI >= 50%. This shows that, in practice, a tumor can be described in several ways and differently (each marker describes a different biological feature).

The best correlation between ultrasound and pathology should be expected in the “expanding type” of tumor growth. This invasion pattern is characterized by a broad front of neoplastic infiltration with a sharp demarcation of tumor tissue from the adjacent healthy tissues. This margin should be clearly identified by ultrasound invasion biomarkers such as MI, TFD, and EMIR. Among these markers, uEMIR seems to be the most subjective and, therefore, the most difficult to evaluate. This characteristic is reflected by multivariate models in which not only uAREA but also uEMIR is missing. However, the last parameter is promising. In physiology, this structure takes part in facilitating sperm transport through modulation of uterine peristalsis and blastocyst implantation; thus, it influences fertility [30]. However, its role in oncology is yet not well elucidated. This intermediate zone between epithelium and muscle layer is lost during EC invasion. Therefore, it can be suggested that uEMIR may be a helpful indicator of early invasion. Our observations did not confirm these assumptions. Perhaps the following suggestion is not strongly supported by the current data, but uEMIR may be a marker of the late invasion of slow-growing tumors (this biomarker proved to be significant in the low-risk group). EMIR assessment was, for example, included in the “REC” (risk of endometrial cancer) scoring system by Dueholm et al. and indicated malignancy in the case of postmenopausal bleeding and endometrial thickness ≥5 mm [36]. Molecular studies seem to confirm the potential role of this intermediate zone in the invasion of cancer that may involve HOX genes [37,38]. Thus far, we have limited knowledge about the role of EMIR assessment in the diagnosis and staging of EC [39].

Tumor volume was included in the scoring systems by Mitamura et al., and Imai et al., who further developed the studies by Todo et al. [40,41,42], although SPE-VOL was measured by MRI and the scoring systems contain a mix of clinical and pathological features. In all these studies, the limit of tumor volume (index) was determined at 36 cm^3^, and our study produced a similar value of 40 cm^3^. Active tumors equate with “high risk”, and indolent tumors with “low risk”. A decrease in LNM risk by tumor volume in a range between 40 and 100 cm^3^ may reflect the point at which an indolent tumor is recognized or may reflect the substantial number of “low-risk” tumors in the study group. A further increase in the risk of metastasis with tumors >100 cm^3^ can be explained by the size of the tumor, which anatomically infiltrates the internal opening of the cervix, thus increasing its access to the lymphatic vessels (Appendix A).

Tumor infiltration is a multidirectional process; the image visible during a 2D ultra-sound examination may not represent a complete infiltration picture (since ultrasound ex-amination is one-dimensional, and histologic sections are multidimensional). In a single TVUS-3D study with the uMI biomarker, not only did Adriano Rodríguez-Trujillo et al. not show the superiority of 3D ultrasound over magnetic resonance imaging in the assessment of infiltration, they also stated that in the case of adenomyosis or uterine fibroids, intraoperative examination of the uterine wall infiltration by cancer was indicated [43]. For these reasons, the uMI, uTFD, uEMIR, and uCSI measurements are observer-dependent and subjective. Therefore, they should be combined in two or more factor models. Moreover, the location of tumors within the uterus may cause discrepancies in the proper evaluation of the ultrasound parameters [44,45]. All issues stated above refer mainly to type I EC. In serious carcinomas, the deepest point of neoplastic infiltration is often synonymous with the deepest location of the lymphatic tumor emboli [46]. Many serious ECs present an image of polypoidal growth only, accompanied by broad peritoneal metastases. However, the most controversial group of cases are endometrioid G3 tumors, which belong to type I EC, but may represent a heterogenic group of cancers histologically, with frequent multiplication of biological features typical of type II [47]. Taking the suggestions stated above into account, a prospective study comprising the analysis of two independent models of uMI:uTFD and uCSI:uSPE-VOL is needed. It would be interesting to know whether the local staging of EC may be enhanced to indicate “high-risk” patients that may benefit from limited or no lymphadenectomy.

Another problem that arises is the potential incorporation of these models into ultrasound machine systems to be able to indicate the risk of LNM in a more applicable way. This requires further research. Validation and application studies are necessary.

## 5. Conclusions

For preoperative ultrasound staging dedicated to LNM risk estimation of endometrial cancer, four parameters are essential, grouped in two pairs: uMI:uTFD and uC-SI:uSPE-VOL. The two-factor model predicts LNM better than the one-factor model. Discretion should be used in choosing a model pair. In the authors’ opinion, the first model is easier to use, because the component parameters, uMI and uTFD, are now used separately.

There are no perfect methods for assessing the invasion of endometrial cancer, and the terminology used is inconsistent. Standardizing the terminology of methods and measurements would allow for better communication between specialists, and perhaps improve the therapeutic qualification for different treatment methods.

## Figures and Tables

**Figure 1 diagnostics-12-02604-f001:**
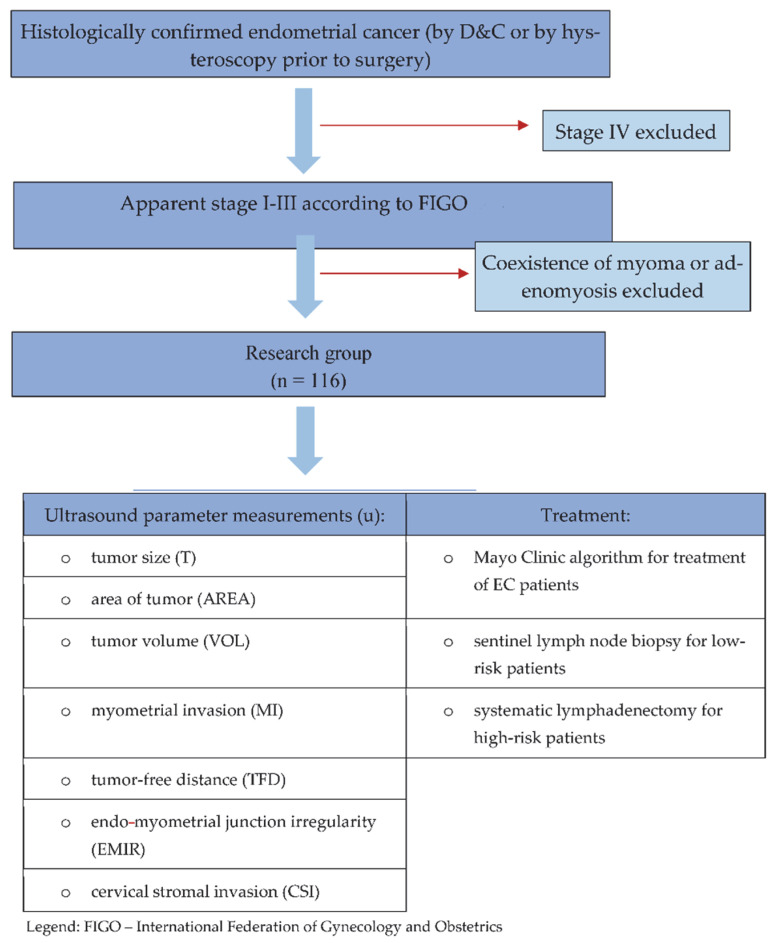
Inclusion and exclusion criteria for study patients, and the flow of the study [17,21]. Mayo Clinic algorithm refres to [22,23].

**Figure 2 diagnostics-12-02604-f002:**
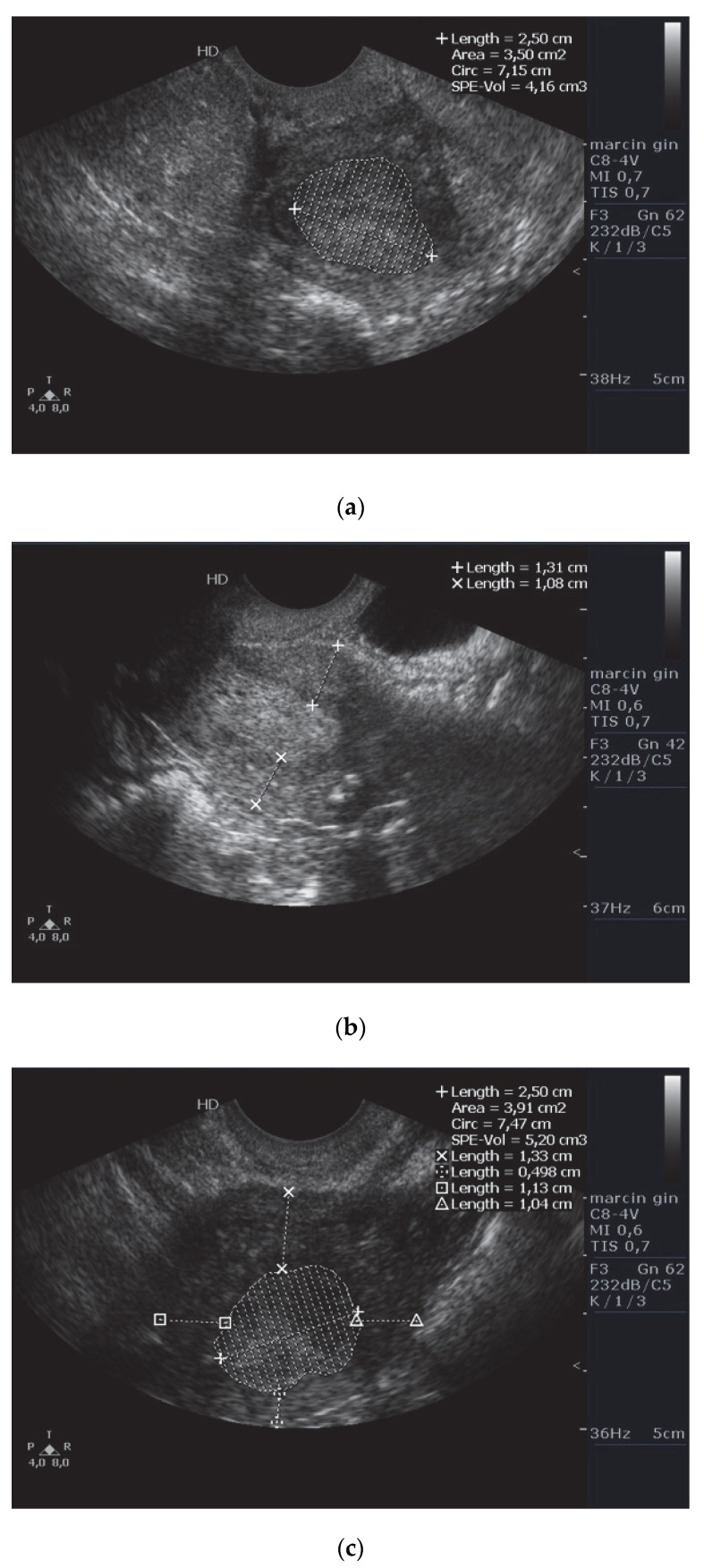
Multiple panels showing how measurements were taken: (**a**) tumor size (T = 2.5 cm), tumor area (“Area” = 3.5 cm^2^), volume (“SPE-Vol” = 4.16 cm^3^); (**b**) myometrial invasion (MI = apparently less than 50%); (**c**) tumor-free distance (TFD = 0.498 cm); (**d**) ruptured endo-myometral junction (lack of endo-myometrium echogenicity strongly suggesting myometrial invasion) (**e**) cervical stromal invasion (CSI).

**Figure 3 diagnostics-12-02604-f003:**
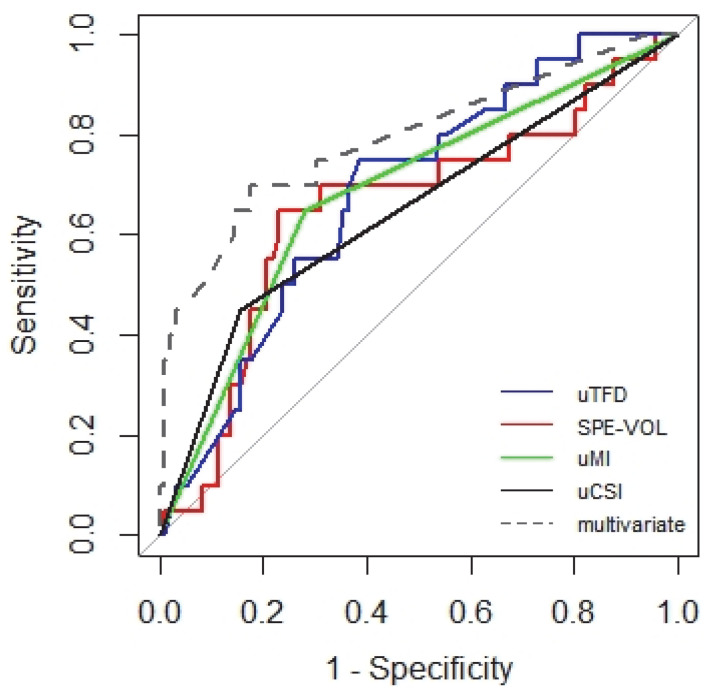
ROC curves showing the most important predictors of lymph node metastasis in a study of 116 women with endometrial cancer.

**Table 1 diagnostics-12-02604-t001:** Baseline characteristics of the study population (116 patients) [18].

Variable	Characteristic	Value
**Age at diagnosis (range)**	Mean +/− SD (range)	63 +/− 8.3 (40–85)
**FIGO stage ***	Number (%)	
Ia		69 (59)
Ib		35 (30)
II		5 (5)
III		7 (6)
**Histologic type**	Number (%)	
Endometrioid		82 (71)
Endometrioid with epithelial differentiation		20 (17)
Serous carcinoma		11 (9)
Carcinosarcoma		3 (3)
**Grade**	Number (%)	
1		41 (36)
2		45 (39)
3		28 (25)
**Lymph node rocedure**	Number (%)	
SLNB only		70 (60)
LND (+SLNB)		46 (40)
**Lymph nodes extracted**	Number	1298
SLNB cases	Number (%)	313 (24)
LND (+SLNB) cases	Number (%)	985 (76)
**Lymph nodes metastases**	Number of patients (%)	20 (17)
**Distribution of positive nodes**	Number (%)	34/1298 (2.62)
Obturator	19 (7 SLN)	
Iliac	13 (2 SLN)	
Para-aortic	2	
**Risk grouping according to initial risk**	Number (%)	Number of patients with metastatic nodes (%)
Low	86 (74)	8 (24)
High	30 (26)	12 (40)

Legend: FIGO—International Federation of Gynecology and Obstetrics, LND—lymphadenectomy, SLN—sentinel lymph node, SLNB—sentinel lymph node biopsy, TVUS—transvaginal ultrasound; * FIGO stage refers to FIGO staging 2009–2018.

**Table 2 diagnostics-12-02604-t002:** Distribution of ultrasound predictors for lymph node metastases in the group of 116 patients with EC.

Ultrasound Variable	Characteristic	Value
T	Number (%)	
≤2 cm		76 (66%)
>2 cm		40 (34%)
AREA [cm^2^]	Mean ± SD (range)	7.49 ± 9.77 (0.161–67)
SPE-VOL [cm^3^]	Mean ± SD (range)	17.00 ± 26.93 (0.033–127)
TFD [mm]	Mean ± SD (range)	7.39 ± 4.83 (0.3–22)
uMI	Number (%)	
<50%		76 (66%)
≥50%		40 (34%)
EMIR	Number (%)	44 (38%)
CSI	Number (%)	24 (20.7%)

Legend: AREA—surface area of tumor; CSI—cervical stromal invasion; EMIR—endomyometrial junction irregularity; MI—myometrial invasion; SPE-VOL—volume of tumor; T—tumor size; TFD—tumor free distance to serosa; Detailed description of the values in the text.

**Table 3 diagnostics-12-02604-t003:** Univariate logit models (models A-J) and multivariate logit model (model K).

Model	Covariate	Est.	Std. Error	*p*-Value	AIC	ACC	AUC (95% CI)	*p*-Value (LRT)
A	(Intercept)	−1.79	0.3	0				
	uSPE-VOL	0.01	0	0.143	108.67	0.75	0.652 (0.507–0.796)	0.159
B	(Intercept)	−1.87	0.32	0	108.18	0.767	0.646 (0.499–0.794)	0.115
	uAREA	0.03	0.02	0.11				
C	(Intercept)	−0.50	0.44	0.254	102.68	0.638	0.683 (0.563–0.803)	0.005
	uTFD	−0.17	0.07	0.012				
D	(Intercept)	−2.68	0.49	0	103.86	0.767	0.689 (0.538–0.840)	
	bs(uSPE-VOL)1	6.81	2.28	0.003				
	bs(uSPE-VOL)2	−3.48	2.94	0.236				0.013
	bs(uSPE-VOL)3	2.36	1.78	0.186				
E	(Intercept)	−2.61	0.61	0	106.23	0.784	0.671 (0.520–0.821)	
	bs(uAREA)1	5.18	4.37	0.215				
	bs(uAREA)2	−0.01	13.98	1				0.038
	bs(uAREA)3	−5.84	38.32	0.878				
F	(Intercept)	−0.62	0.93	0.506	106.32	0.638	0.683 (0.563–0.803)	
	bs(uTFD)1	−1.37	4.14	0.74				
	bs(uTFD)2	−0.58	6.49	0.929				0.04
	bs(uTFD)3	−8.04	11.92	0.5				
G	(Intercept)	−2.29	0.4	0	101.17	0.707	0.684 (0.568–0.801)	
	uMI	1.56	0.52	0.003				0.002
H	(Intercept)	−2.23	0.4	0	103.34	0.672	0.664 (0.547–0.781)	
	uEMIR	1.36	0.52	0.009				0.007
I	(Intercept)	−2.00	0.32	0	103.11	0.776	0.647 (0.529–0.765)	
	uCSI	1.49	0.53	0.005				0.06
J	(Intercept)	−3.43	0.82	0	104.02	0.69	0.654 (0.535–0.773)	0.01
	Size	1.29	0.51	0.011				
K	(Intercept)	−2.62	0.44	0	94.81	0.802	0.791 (0.673–0.91)	
	uMI:bs(uTFD)1	−13.24	4.74	0.005				
	uMI:bs(uTFD)2	−54.62	27.47	0.046				
	uMI:bs(uTFD)3	121.59	60.64	0.044				0.005
	uCSI:bs(uSPE-VOL)1	10.09	3.51	0.004				0.006
	uCSI:bs(uSPE-VOL)2	−15.04	9.3	0.105				
	uCSI:bs(uSPE-VOL)3	9.39	9.07	0.301				

Legend: ACC—accuracy, AIC—Akaike Information Criterion, AREA—surface area of tumor; AUC—area under the ROC curve; bs—b-spline; CSI—cervical stromal invasion; EMIR—endomyometrial junction irregularity; LRT—likelihood ratio test; MI—myometrial invasion; SPE-VOL—volume of tumor; T—tumor size; TFD—tumor free distance to serosa; u—ultrasonographic; Detailed description of the values in the text.

**Table 4 diagnostics-12-02604-t004:** Univariate logit models (models A′–J′) and multivariate logit model (K′) for high-risk group (*n* = 31).

Model	Covariate	Est.	Std. Error	*p*-Value	AIC	ACC	AUC (95% CI)	*p*-Value (LRT)
A′	(Intercept) uSPE-VOL	−0.13 −0.01	0.48 0.01	0.786 0.509	49.62	0.618	0.479 (0.268–0.689)	0.501
B’	(Intercept) uAREA	−0.12 −0.01	0.53 0.03	0.827 0.550	49.69	0.618	0.475 (0.265–0.685)	0.536
C′	(Intercept) uTFD	0.12 −0.09	0.58 0.08	0.836 0.316	49.01	0.588	0.571 (0.375–0.768)	0.304
D′	(Intercept)	−1.04	0.77	0.176	50.22	0.765	0.686 (0.483–0.889)	
	bs(uSPE.VOL)1	5.11	3.12	0.101				
	bs(uSPE.VOL)2	−4.68	3.24	0.148				0.278
	bs(uSPE.VOL)3	0.87	1.91	0.650				
E′	(Intercept)	−0.84	0.82	0.307	52.09	0.765	0.643 (0.423–0.863)	
	bs(uAREA)1	2.863.28	3.84	0.394				
	bs(uAREA)2	−3.11	6.09	0.610				0.577
	bs(uAREA)3	−3.58	9.19	0.697				
F′	(Intercept)	−0.14	0.77	0.855	52.57	0.618	0.600 (0.404–0.796)	
	bs(uTFD)1	0.05	3.42	0.989				
	bs(uTFD)2	0.30	3.56	0.933				0.683
	bs(uTFD)3	−2.51	3.49	0.472				
G′	(Intercept)	−0.81	0.60	0.177	49.11	0.559	0.582 (0.416–0.748)	
	uMI	0.72	0.74	0.335				0.328
H′	(Intercept)	−0.69	0.55	0.206	49.38	0.559	0.571 (0.399–0.744)	
	uEMIR	0.59	0.71	0.411				0.407
I′	(Intercept)	−0.69	0.55	0.206	49.38	0.559	0.571 (0.399–0.743)	
	uCSI	0.59	0.71	0.411				0.407
J′	(Intercept) Size	−0.48 0.07	1.36 0.77	0.726 0.928	50.06	0.471	0.507 (0.347–0.667)	0.928
K′	(Intercept)	−0.89	0.45	0.048	45.19	0.706	0.675 (0.517–0.833)	.
	uCSI:uMI	1.74	0.82	0.035				0.027

## Data Availability

The collected clinical material is in electronic form in the Clinic’s repository and may be made available in an anonymized form upon request.

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
