# Peer review of "Incorporation of Tumor-Free Distance and Other Alternative Ultrasound Biomarkers into a Myometrial Invasion-Based Model Better Predicts Lymph Node Metastasis in Endometrial Cancer: Evidence and Future Prospects"

_diagnostics, 2022, doi:10.3390/diagnostics12112604_

Round 1
Reviewer 1 Report
Interesting article on the ultrasound prediction of LNM in endometrial cancer. It provides interesting markers of LNM. However, some comments are due:
Title
I would incorporate the tumor free distance in the title
Abstract
- in the abstract I would add the objective of the study
Intro
- Line 43: I would replace the word “before” as US scan is part of the preop staging even after diagnosis is made
- Line 43: I would suggest removing the statement starting with “The task…”
- Line 46: I would be careful stating that “US is a diagnostic method” as endometrial cancer is not diagnosed with US but with biopsy; US is useful to determine the pre-operative tumor extension
- Line 59: I would suggest rephrasing this sentence as not only node positive EC are high risk (see Concin et al. ESGO guidelines, IJGC)
- Line 66-69: I would suggest removing covid 19 referral
Methods
- Despite s previous study was published, inclusion and exclusion criteria should be reported in the present study as well
- In the flow chart I would add “apparent” FIGO I-III as final figo stage is post surgery
- Can the authors explain why they did include figo stage III as well? Which information could add the knowledge of lymph node status ?
- What do the Authors mean with “regional” lymphadenectomy (Line 146)
- Line 146: what do the authors mean with “diagnosed prognostic factors for lymph node recurrence”? I would suggest clarifying this statement
- Can the authors report the evidence for performing SLN biopsy in patients with comorbidities?
Results
- a table with clinico-pathological characteristics of the included patients should be reported (including patients considered at low and high risk)
- Line 185: I would replace “neoplastic metastases” with “lymph node metastasis”
- What do the authors mean with “bs”?
- The results and the fact that the combination of uTFD and other parameters could predict LNM better than uMI alone, should be more clearly explained
English grammar needs to be improved
Author Response
Please find enclosed document.

Reviewer 2 Report
The researchers evaluated alternative ultrasound biomarkers for prediction of lymph node metastasis in endometrial cancer and propose two models. The manuscript is well written, well structed and the methods and details of the study are well described. I would like only some comments to make.
Minor suggestions:
Line 19: “who had received 2D transvaginal….” Please revised. The transvaginal ultrasonographic examination performed on the patients.
Line 38: I don't understand why it's included in the keywords the COVID-19.
Line 65-68: I think there is no reason to report in COVID-19.
Line 115 - 2.2. Ultrasound examination: since the study evaluates indicators resulting from ultrasonographic examination, I believe that the basic settings of the ultrasound equipment with which the measurements were made should be mentioned, e.g. frequency, depth and gray map.
Line 117: “vaginal probe…”, I suggest “transvaginal transducer”.
Line 136: “ultrasound researcher” please check the terminology.
Line 141 - fig(d): because color Doppler has been used, there should be a reference-commentary of this methodology in materials and methods.
Table 1: Number(T) – in false line.
Figure 3: too small figure.
Please revised cm3 in lines 224,303,304,306,308.
Author Response
Please find enclosed document.

Round 2
Reviewer 1 Report
Thanks for the changes provided.
Author Response
Thank you for your revision again.
Due to the fact that the Reviewer did not leave any further comments, we only made minor stylistic corrections in the text.
We hope that the linguists will eventually help improve the text.